# Diet Rich in Simple Sugars Promotes Pro-Inflammatory Response via Gut Microbiota Alteration and TLR4 Signaling

**DOI:** 10.3390/cells9122701

**Published:** 2020-12-16

**Authors:** Alena Fajstova, Natalie Galanova, Stepan Coufal, Jana Malkova, Martin Kostovcik, Martina Cermakova, Helena Pelantova, Marek Kuzma, Blanka Sediva, Tomas Hudcovic, Tomas Hrncir, Helena Tlaskalova-Hogenova, Miloslav Kverka, Klara Kostovcikova

**Affiliations:** 1Laboratory of Cellular and Molecular Immunology, Institute of Microbiology of the Czech Academy of Sciences, Videnska 1083, 142 20 Prague, Czech Republic; alena.fajstova@biomed.cas.cz (A.F.); natalie.galanova@biomed.cas.cz (N.G.); coufal@biomed.cas.cz (S.C.); janca.malku@gmail.com (J.M.); tlaskalo@biomed.cas.cz (H.T.-H.); kverka@biomed.cas.cz (M.K.); 2Department of Cell Biology, Faculty of Science, Charles University, Vinicna 7, 128 01 Prague, Czech Republic; 3Laboratory of Fungal Genomics and Metabolism, Institute of Microbiology of the Czech Academy of Sciences, Videnska 1083, 142 20 Prague, Czech Republic; kostovci@biomed.cas.cz; 4BIOCEV, Institute of Microbiology of the Czech Academy of Sciences, Prumyslova 595, 252 42 Vestec, Czech Republic; 5Department of Genetics and Microbiology, Faculty of Science, Charles University, Vinicna 5, 128 44 Prague, Czech Republic; 6Laboratory of Molecular Structure Characterization, Institute of Microbiology of the Czech Academy of Sciences, Videnska 1083, 142 20 Prague, Czech Republic; martina.buganova@biomed.cas.cz (M.C.); pelantova@biomed.cas.cz (H.P.); kuzma@biomed.cas.cz (M.K.); sediva@rek.zcu.cz (B.S.); 7Department of Analytical Chemistry, Palacky University, 17. Listopadu 1192/12, 771 46 Olomouc, Czech Republic; 8Faculty of Applied Sciences, University of West Bohemia, Univerzitni 8, 301 00 Plzen, Czech Republic; 9Laboratory of Gnotobiology, Institute of Microbiology of the Czech Academy of Sciences, Doly 183, 549 22 Novy Hradek, Czech Republic; hudcovic@biomed.cas.cz (T.H.); hrncir@biomed.cas.cz (T.H.)

**Keywords:** inflammatory bowel diseases, neutrophils, metabolites, microbiome, mucosal barrier, high-sugar diet

## Abstract

Diet is a strong modifier of microbiome and mucosal microenvironment in the gut. Recently, components of western-type diets have been associated with metabolic and immune diseases. Here, we studied how high-sugar diet (HSD) consumption influences gut mucosal barrier and immune response under steady state conditions and in a mouse model of acute colitis. We found that HSD significantly increased gut permeability, spleen weight, and neutrophil levels in spleens of healthy mice. Subsequent dextran sodium sulfate administration led to severe colitis. In colon, HSD significantly promoted neutrophil infiltration and increased the levels of IL-6, IL-1β, and TNF-α. Moreover, HSD-fed mice had significantly higher abundance of pathobionts, such as *Escherichia coli* and *Candida*, in fecal samples. Although germ-free mice colonized with microbiota of conventionally reared mice that consumed different diets had equally severe colitis, mice colonized with HSD microbiota showed markedly increased infiltration of neutrophils to the gut. The induction of colitis in Toll-like receptor 4 (TLR4)-deficient HSD-fed mice led to significantly milder colitis than in wild-type mice. In conclusion, our results suggested a significant role of HSD in disruption of barrier integrity and balanced mucosal and systemic immune response. In addition, these processes seemed to be highly influenced by resident potentially pathogenic microbiota or metabolites via the TLR4 signaling pathway.

## 1. Introduction

Saccharides constitute the greatest proportion of our daily food intake. Today’s high consumption of simple refined sugars and lack of complex polysaccharides, such as fiber, are typical features of westernized diets [1]. In the last decades, the consumption of foods rich in simple or added sugars grew not only in high-income countries, but also in middle- and low-income countries [2]. Recent research on the effect of high consumption of simple refined sugars has shown many potentially detrimental effects on human health, including metabolic, gastrointestinal, cardiovascular, and neurological diseases [3,4,5]. Studies in various animal models have revealed that high sugar intake influences the development of allergy [6], hypertension, and insulin resistance [7] and causes cognitive decline [8,9,10,11] and thus can affect the whole organism, for instance, by increasing oxidative stress [12,13,14].

Diet is a strong modulator of gut microbiome and related mucosal immune responses [15,16]. High amounts of dietary simple sugars decrease microbial diversity and lead to the depletion of luminal short-chain fatty acids (SCFAs) [17,18]. SCFAs have immunomodulatory properties which can influence the recruitment of colonic regulatory T cells and antimicrobial activity of macrophages [19,20]. Thus, changes in intestinal microbiota composition and metabolism are rapidly influencing gut mucosal immune system response and vice versa [15]. To maintain homeostasis, the presence of commensal gut microbiota and mucus layer is crucial, as it prevents invasion and adherence of pathogenic microorganisms and helps to maintain intestinal barrier integrity [21]. Impaired barrier, that is, increased intestinal permeability, enables the translocation of microbes and their components, such as lipopolysaccharide (LPS) and mannan. These microbial components are then recognized through various pattern recognition receptors, including toll-like receptors, NOD-like receptors, and C-type lectin receptors [21]. Most of them activate the NF-κB pathway, which increases pro-inflammatory response and the production of cytokines and chemokines, thereby enhancing the recruitment of neutrophils and monocytes to the site of inflammation [22,23]. Thus, continuous mild stimulation of pattern recognition receptors (for instance, with LPS) accompanied by barrier failure can trigger systemic low-grade inflammation [24,25]. This has been observed in subjects consuming high-fat diet, which resulted in dysbiosis and increased serum LPS levels [26].

Although the influence of westernized diets on health has been reported by numerous studies, most of them were clinical trials assessing the consequences of westernized diet consumption and possible effects on human health [27,28,29]. The data about underlying precise mechanisms are mostly missing as well as the influence of certain dietary components. There are clinical studies evaluating the effect of high intake of simple sugars in inflammatory bowel disease (IBD) patients, but the results are inconclusive [30,31]. Unfortunately, the discrepancies are also found in animal studies; nevertheless, most of them agree that high intake of simple sugars is detrimental in mice with colitis [17,32,33,34]. Moreover, the effect of a high-sugar diet alone on immune system reactivity has been studied only marginally. In the present study, we focused on the influence of a high-sucrose diet (HSD) and a high-fiber diet (HFiD) on intestinal barrier function, interaction with gut microbiota, and mucosal and systemic immune response in healthy mice and in mouse models of acute and chronic colitis.

## 2. Materials and Methods

### 2.1. Mice, Experimental Diets, and Gut Permeability

Six- to eight-week-old wild-type BALB/c female mice were reared conventionally or under germ-free (GF) conditions. For the study of underlying mechanisms, we used Recombination activating gene 2 protein knockout mice (RAG2^−/−^)- and Toll-like receptor 4 (TLR4)- deficient mice on BALB/c background. All mice were obtained from the breeding colonies of the Institute of Microbiology of the Czech Academy of Sciences. Prior to the experiments, mice were fed with a Maintenance diet for rats and mice (SD; Cat# 1324). At the beginning of the experiments, mice were given either a control diet (CD; Cat# C 1000), a carbohydrate-rich diet (HSD; Cat# C 1010), or a crude fiber-rich diet (HFiD; Cat# C 1014). All diets were from Altromin Spezialfutter GmbH & Co. KG, Lage, Germany. The main components of the diets are summarized in Appendix A and saccharide content in Appendix A. Mice were kept on specific diets at least 3 weeks before colitis induction. Prior and during the experiments, mice were fed ad libitum, had free access to clean tap water, and were housed in standard conditions. For microbiota transfer to germ-free mice, we used donor wild-type BALB/c mice that were fed CD, HSD, or HFiD. After three weeks, mice were sacrificed and intestinal content from colon and ileum was collected in separate tubes, pooled for all mice in respective groups, and diluted in 2 mL of sterile phosphate-buffered saline (PBS). Immediately, 200 µL of the solution was administered to the recipient GF wild-type BALB/c mice (exGF), and ileal content via oral gavage and colonic content via rectal gavage were administered with a steel bulb-tipped gavage needle. After the transfer, mice were further given sterile water and CD and were kept in individually ventilated cages for two weeks before colitis induction. The study was carried out in accordance with the recommendations of the ethics standards defined by the European Union legislation on the use of experimental animals (2010/63/EU) and Czech animal welfare act. All mice were used according to the procedures approved by the Institute of Microbiology Animal Care and Use Committee (approval IDs: 108/2016 and 109/2019).

To study intestinal permeability, each animal received orally 440 mg/kg of fluorescein isothiocyanate (FITC)-labeled dextran (3–5 kDa; Merck KGaA, Darmstadt, Germany; Cat# FD4), and after 4 h, we measured fluorescence in serum, as published previously by us [35]. The concentration of mouse lipopolysaccharide-binding protein (LBP) in serum was measured by Mouse LBP ELISA (Hycult Biotech, Uden, The Netherlands; Cat# HK205-02), which is certified for serum samples. The assay was performed according to the manufacturer’s instruction.

### 2.2. Experimental Mouse Models

To induce colitis, mice were given 3% dextran sulfate sodium (DSS, 36–50 kDa; MP Biomedicals, CA, USA; Cat# 02160110) solution in autoclaved drinking water. In the case of acute colitis, DSS was administered for 6–8 days (6 days for RAG2^−/−^ mice and exGF mice, 8 days for all other experiments). The treatment scheme is shown in Appendix A. To induce chronic colitis, mice were given 3% DSS in three cycles, each cycle consisting of five days of 3% DSS, followed by nine days of tap water. After this period, the experiment was terminated and severity of colitis was evaluated. Final mouse weight, colon length, and spleen weight were measured and disease activity index (DAI) was calculated as a mean of weight loss, stool consistency, and rectal/occult bleeding, as previously published [36]. Pieces (1 cm long) of distal colon were collected and fixed in 4% formalin, dehydrated in ethanol, and embedded in paraffin. Colonic sections (4 μm) were rehydrated and stained by hematoxylin and eosin. Subsequent microscopic evaluation was made by an experienced pathologist in a blinded manner. The damage of the mucosa was scored according to the degree of leukocyte infiltration into lamina propria and submucosa and the extent of ulceration. The final score ranged from 0 (no signs of colitis) to 3 (severe colitis) [35].

To observe the effect of diet on inflammatory processes outside the gut, we used carrageenan-induced footpad paw edema, mediating the transient non-T-cell inflammatory response. The mice were anesthetized by ketamine-xylazine anesthesia (Bioveta a.s., Ivanovice na Hane, Czech Republic). The thickness of both the hind footpads was measured. Then, 25 µL of 2.5% λ-carrageenan (Merck; Cat# 22049) was injected in the plantar side of the right hind footpad. Sterile saline was injected in the plantar side of left hind footpad. After 24 h, the experiment was terminated and footpad thickness was measured and swelling response was calculated by subtracting the footpad thickness measured before injection from that measured after 24 h.

### 2.3. Cell Suspension Preparation and Flow Cytometry

Mouse spleens and mesenteric lymph nodes were collected upon experiment termination and mashed separately. Cell suspensions were than filtered with a 70 µm cell strainer (Becton Dickinson, Prague, Czech Republic; Cat# 352350). Spleen suspensions were than washed, centrifuged (300× *g*, 5 min, 4 °C), and treated with red blood cells lysis buffer (1 mM EDTA, 150 mM NH_4_Cl, 10 mM KHCO_3_). After 5 min of incubation, the tubes were centrifuged, supernatant was discarded, and cells were diluted in complete RPMI medium (Merck; Cat# R0883) containing 10% heat-inactivated fetal bovine serum (FBS; Biochrom GmbH, Berlin, Germany; Cat# S 0115) and 1% antibiotic-antimycotic solution (Merck; Cat# P4333). Finally, cell suspensions were counted and diluted to the concentration of 2 × 10^6^ cells/mL. Colon samples of about 3 cm long were collected and processed according to the published protocol [37].

Cell suspensions were used for measurement of subsets of T cells and mononuclear cells by flow cytometry. Briefly, cells were blocked with 10% of normal mouse serum in PBS. Next, the cells were stained with fluorochrome-conjugated antibodies recognizing extracellular epitopes and Fixable Viability Dye eFluor 780 (Thermo Fisher Scientific, Waltham, MA, USA; Cat# 65-0865-18) to distinguish live and dead cells (the list of used antibodies is in Appendix A). Before staining for intracellular antigens, the cells were fixed and permeabilized with eBioscience Foxp3/Transcription Factor Staining Buffer Set (Thermo Fisher Scientific; Cat# 00-5523-00). Samples were measured on LSR II (BD Biosciences, CA, USA) and data were analyzed with the FlowJo software (Tree Star Inc., Ashland, OR, USA; RRID: SCR_008520). Gating strategies for immune cell analyses are shown in Appendix A.

### 2.4. Cell Cultivation, Tissue Cultures, and Cytokine Measurement

Plates were treated with the Ultra-LEAF^TM^ purified anti-mouse CD3 antibody (5 μg/mL; clone 17A2; Biolegend, San Diego, CA, USA; Cat# 100238), and then, the cell suspensions from mesenteric lymph nodes and spleens were distributed to the plate at a concentration 2 × 10^6^ cells/mL per well. Finally, the Ultra-LEAF^TM^ purified anti-mouse CD28 antibody (4 μg/mL; clone E18; Biolegend, Cat# 122022) was added and the plate was incubated in a humidified incubator (37 °C, 5% CO_2_) for 48 h. Then, supernatants were collected and frozen at −20 °C until analysis.

Colon biopsies were weighted (approximately 3–7 mg) and cultivated in 500 µL of complete RPMI medium in a humidified incubator (37 °C, 5% CO_2_) for 48 h. Supernatants were collected and frozen at −20 °C.

Cytokines in supernatants were measured using appropriate DuoSet ELISA Development Systems (Bio-Techne, Minneapolis, MN, USA; Cat# DY410, DY406, DY401, and DY421) according to the manufacturer’s instructions.

### 2.5. Isolation of Total RNA, Reverse Transcription, and qPCR

Colon and ileum biopsies were collected when the experiment was terminated. The tissues were weighted and immediately immersed in RNAprotect Tissue Reagent (Qiagen, Hilden, Germany; Cat# 76106) and stored at −20 °C. Samples were transferred to Lysing Matrix D tubes (MP Biomedicals, Santa Ana, CA, USA; Cat# 116913050) and homogenized in FastPrep-24 (MP Biomedicals). Total RNA was isolated using TRI Reagent (Zymo Research, Irvine, CA, USA; Cat# R2050). Next, the samples were treated with DNAse (TURBO DNA-*free* Kit; Thermo Fisher Scientific; Cat# AM1907). cDNA was synthesized (using total 500 ng RNA) with a SuperScript IV Reverse Transcriptase kit (Thermo Fisher Scientific; Cat# 10777019) and Oligo(dT)_12–18_ Primer (Thermo Fisher Scientific; Cat# 18418012). The qPCR reaction was performed using gb SG PCR Master Mix (Generi Biotech s.r.o., Hradec Kralove, Czech Republic; Cat# 3005) and specific primers (Generi Biotech, Appendix A). The qPCR runs were performed on CFX384 Touch (Bio-Rad, Hercules, CA, USA) and LightCycler 480 Real-Time PCR System (Roche s.r.o., Prague, Czech Republic) machines. The cycling parameters were as follows: 3 min at 95 °C, 40 cycles of 30 s at 94 °C, 40 s at 60 °C, and 60 s at 72 °C. Data were normalized to expression levels of the reference genes (ribosomal protein S12 and eukaryotic elongation factor 2) and the relative fold change was calculated with 2^−ΔΔCt^ method.

### 2.6. Microbiome Analysis

Stool samples were collected at three different time points: before switching to experimental diets (day −21), before disease induction (day 0), and at the termination of the experiment (day 7). Total DNA was extracted using MasterPure Complete DNA and RNA Purification Kit (Epicenter, Illumina Inc., Madison, WI, USA; Cat# MC85200) with repeated bead beating in Lysing Matrix Y tubes (Cat# 116960500) using the FastPrep homogenizer (both were from MP Biomedicals) and PCR inhibitors were removed using InhibitEx Tablets (Qiagen; Cat# 19590). DNA was then standardized using a Qubit dsDNA High Sensitivity kit (Thermo Fisher Scientific). PCR targeting V3 and V4 regions of bacterial 16S was conducted using the Kapa HiFi DNA polymerase (Kapa Biosystems, Wilmington, MA, USA; Cat# KK2602 ), primers 341F (5′-CCTACGGGNGGCWGCAG-3′) and 806R (5′-GGACTACHVGGGTWTCTAAT-3′) [38] (Generi Biotech), and 10% bovine serum albumin (BSA; Merck; Cat# 8894). Cycling conditions consisted of initial denaturation (94 °C, 3 min) followed by 30 cycles of denaturation (94 °C, 30 s), annealing (54.2 °C, 45 s), extension (72 °C, 75 s), and final extension (72 °C, 10 min). For PCR targeting fungal internal transcribed spacer 1 (ITS1) region, PPP Master Mix (Top-Bio, Vestec, Czech Republic; Cat# P126) with 18SF (5′-GTAAAAGTCGTAACAAGGTTTC-3′) and 5.8SR (5′-GTTCAAAGAYTCGATGATTCAC-3′) primers and 10% BSA was used [39]. Cycling conditions were 95 °C, 5 min; 35 cycles of 95 °C, 30 s; 50 °C, 30 s; 72 °C, 60 s; and 72 °C, 10 min. PCR triplicates were pooled and purified by SequalPrep Normalization Plate Kit (Thermo Fisher Scientific; Cat# A1051001). Samples within library were pooled and sequencing adaptors were ligated using the TruSeq DNA PCR-free LT Sample Preparation Kit (Illumina, Madison, WI, USA; Cat# FC-121-3001). Ligated libraries were quantified with KAPA Library Quantification Kit (Kapa Biosystems, Wilmington, MA, USA; Cat# KK4824) and sequenced on MiSeq Illumina Platform using Miseq Reagent Kit v3 (Illumina; Cat# MS-102-3003) at The Genomics Core Facility, CEITEC (Brno, Czech Republic). Sequencing data were processed using QIIME version 1.9.1 [40]. Quality filtering, chimera detection, read demultiplexing, and read clustering were done as described previously [41]. Fungal reads were in addition extracted for the ITS1 region using the ITSx package [42]. The identification of representative sequences was done using the Ribosomal Database Project (RPD) classifier [43] against bacterial GREENGENES database 13.8 [44] and fungal UNITE database 7.2 (UNITE Community (2017): UNITE QIIME release (version 01.12.2017; UNITE Community; https://doi.org/10.15156/BIO/587481). Finally, the operational taxonomic units (OTU) table was produced. The data are available in the Sequence Read Archive (SRA) at http://www.ncbi.nlm.nih.gov/sra under the submission number SUB8395994. Briefly, for microbiota analysis, Chao1 index was used to describe alpha diversity, and principal co-ordinate analysis (PCoA) based on Bray–Curtis dissimilarity metrics was used to describe beta diversity. Next, linear discriminant analysis effect size (LEfSe; RRID: SCR_014609) was used to determine the features discriminating communities in each group [40,45]. The functional composition of a bacterial metagenome was predicted by the Phylogenetic Investigation of Communities by Reconstruction of Unobserved States (PICRUSt) tool, using the 16S amplicon data [39].

### 2.7. Metabolome Analysis

Samples for nuclear magnetic resonance (NMR)-based metabolomics were prepared by the extraction of fecal pellets, collected at day 0, with phosphate-buffered saline (pH 7.4), as described previously [36]. The NMR experiments were performed on a Bruker Avance III 600 MHz spectrometer (Bruker BioSpin, Rheinstetten, Germany) equipped with a 5 mm TCI cryoprobe. The metabolomic analysis was based on ^1^H NMR spectra acquired using the Carr–Purcell–Meiboom–Gill (CPMG) pulse sequence with water pre-saturation during relaxation delay d1 = 4 s (number of scans (NS) = 256, 64 k of data points (TD), spectral width (SW) = 20 ppm). Additional *J*-resolved experiments with pre-saturation were performed to facilitate metabolite identification (NS = 4, SW = 16 ppm, TD = 8 k, number of increments =40, SW = 78.125 Hz in the indirect dimension, d1 = 2 s). CPMG spectra were pre-processed using the NMRProcFlow software [46]. All spectra were baseline corrected, aligned, and divided into bins by intelligent bucketing. Binned data were submitted to the MetaboAnalyst 4.0 software [47] and Matlab software (MATLAB version 8.6; The MathWorks Inc., Natick, MA, USA) for statistical evaluation. Next, binned spectra were normalized by the probabilistic quotient normalization (PQN) method [48] with a pooled CD group as a reference.

Pareto-scaled data were subjected to principal component analysis (PCA) to visualize the grouping and to detect outliers. Afterward, the partial least square discriminant analysis (PLS-DA) was executed to identify bins contributing to the diet-induced group separation. The quality of the PLS-DA model was determined by the leave-one-out cross-validation and by the permutation test. The binned spectra were also examined using univariate statistics. Based on the results of the Lilliefors test for normality, analysis of individual bins was achieved by parametric one-way analysis of variance (ANOVA). Significantly changed bins were assigned to metabolites using the Chenomx software (Chenomx Inc., Edmonton, AB, Canada) and by comparison with the Human Metabolome Database (www.hmdb.ca) or with the previously published spectral data.

### 2.8. Statistical Analysis

Data were analyzed using GraphPad Prism version 8.0.0 for Windows (GraphPad Software, San Diego, CA, USA; www.graphpad.com). Statistical differences between two groups were calculated by unpaired *t* test. In other cases, one-way analysis of variance (ANOVA) with Dunnett’s multiple comparison test or two-way ANOVA with Bonferroni post hoc test were used. Data were expressed as mean ± standard deviation (SD) unless otherwise stated. Values of *p* < 0.05 were considered significantly different.

## 3. Results

### 3.1. Diet Rich in Simple Carbohydrates Promotes Pro-Inflammatory Tuning

When administered to healthy mice, the diet rich in simple carbohydrates increased intestinal permeability and spleen size, without macroscopically or microscopically damaged colon mucosa (Figure 1A). Compared with CD-fed mice, mice fed with HSD had generally higher state of immune system activation, as demonstrated by increased proportion of both neutrophils and Th17 cells in the spleen or increased proportions of both resident (Ly6C^low^) and pro-inflammatory (Ly6C^high^) macrophages in the colon tissue (Figure 1B). While HSD did not change pro-inflammatory cytokine production by anti-CD3/anti-CD28-stimulated cells from spleen or mesenteric lymph nodes (mLN) (Figure 1C), it increased *Il1b* expression in colon tissue (Figure 1D). Complex plant polysaccharides in HFiD did not have this pro-inflammatory effect and they even decreased the proportion of neutrophils and anti-CD3/anti-CD28-stimulated production of TNF-α in the spleen (Figure 1B,C). Neither diet significantly changed the expression or production of other cytokines in the colon tissue, although there was a trend to higher expression of *Nos2* and *Tnfa* in HSD-fed mice (Figure 1D and Appendix A). Together, these results suggested increased pro-inflammatory tuning in healthy HSD mice.

### 3.2. Diet Rich in Simple Sugars Worsens Acute DSS Colitis in a T-Cell-Independent Manner

Next, to analyze if barrier failure and pro-inflammatory tuning by the HSD had consequences for the pathological inflammatory process, we induced acute colitis in HSD- and HFiD-fed mice. We found that, indeed, mice fed with HSD were significantly more sensitive to acute colitis as compared with either CD- or HFiD-fed mice (Figure 2A). Severe colitis in HSD-fed mice was accompanied by a massive increase in neutrophils in colon tissue (Figure 2B), increased anti-CD3/anti-CD28-stimulated production of IL-17 in mesenteric lymph nodes (Figure 2C), and increased *Tnfa* expression (Figure 2D) and IL-6 and IL-1β production (Figure 2E) in the colonic tissue. Similar to healthy mice, HSD-fed mice showed a slight and non-significant increase in colonic expression of pro-inflammatory markers *Nos2* and *Il17f* (Appendix A). The severity of acute colitis in HFiD-fed mice was only slightly milder as compared with control mice, but it was accompanied with a decrease in colonic *Il1b* expression (Figure 2D), suggesting lower pro-inflammatory tuning, which we had already observed in HFiD-fed healthy mice.

Even if only minor changes in T cells were observed in HSD-fed mice, under both healthy and colitic conditions, we induced acute colitis in RAG2^−/−^ mice fed with different diets. We found that, even in the absence of the adaptive immune system, mice fed with HSD had more severe colitis and mice fed with HFiD had milder colitis as compared with CD-fed mice (Figure 3). This suggested that the pro-inflammatory effects of HSD were not mediated by adaptive immunity. Moreover, healthy HSD-fed RAG2^−/−^ mice showed increased intestinal permeability (Appendix A) and increased percentage of neutrophils in spleen (Appendix A), suggesting that adaptive immune response was not responsible for the pro-inflammatory tuning described in immunocompetent BALB/c mice.

### 3.3. Diet Rich in Simple Carbohydrates Increases Severity of Chronic Colitis

Next, we induced chronic colitis in BALB/c mice by three cycles of DSS, which mimicked alternation of relapses and remissions typical for chronic human IBD. Similar to acute colitis, mice fed with HSD had significantly more severe colitis during the whole experiment as compared with CD- or HFiD-fed mice (Figure 4A). In addition, mice fed with HSD had increased proportion of neutrophils in their spleen and macrophages infiltrating their colon tissue had higher production of inducible nitric oxid synthase (iNOS), suggesting more severe systemic and local inflammatory response (Figure 4B). Interestingly, compared with controls, mice fed with HFiD had milder damage to colon, as measured by colon shortening, and lower proportions of pro-inflammatory cells (e.g., neutrophils, macrophages, Th17, and innate lymphoid cells type 3) in the colon tissue (Figure 4B). This anti-inflammatory effect of HFiD was confirmed by a significant decrease in IL-6 and TNF-α production in the colon tissue (Figure 4C), but pro-inflammatory gene expression was not decreased significantly (Figure 4D), whereas HSD led to slightly increased expression (Figure 4C,D).

### 3.4. HSD Changes the Expression of Gut Barrier-Associated Genes in Colon

Since HSD induced changes in intestinal permeability and mucosal immune response in healthy mice, we analyzed the mRNA expression of genes related to gut barrier function in the colon of healthy mice. Compared with CD-fed mice, HSD-fed mice had reduced *Muc2* and increased *Il22* mRNA expression. This pattern was accompanied with significantly increased expression of antimicrobial peptides genes *Reg3b* and *Reg3g.* The expression of *Clec7a* (dectin 1) and *Clec4n* (dectin 2), *Nfkbiz*, *Tjp1*, and *Ocln* was increased in healthy mice fed with HSD vs. CD, whereas *Tlr4* was not changed (Figure 5A and Appendix A). When colitis was induced, in all groups, the clear expression pattern of *Muc2*, *Reg3b*, and *Reg3g* was lost, but *Clec7a, Clec4n*, and *Il22* mRNA expression was still increased in HSD- vs. CD-fed mice (Figure 5B and Appendix A). In addition, gene expression of *Tjp1*, *Tlr4* and its negative regulator *Irak3*, but not *Nfkbiz*, was slightly increased and *Ocln* was less expressed in HSD-fed colitic mice (Figure 5B and Appendix A).

### 3.5. Diet Significantly Influenced Gut Bacterial and Fungal Microbiota Composition

To observe whether diet consumption and changed expression of antimicrobial proteins influenced gut microbiota composition, we collected fecal samples on days −21, 0, and 7 and processed them for sequencing. Alpha diversity indices for bacteria mostly showed a non-significant drop after transfer to special diet and even greater decrease after DSS colitis induction. This reduction was partly prevented in HFiD-fed mice (Figure 6A), suggesting protection of the microbial ecosystem. Diet change led to marked shifts in the composition of gut bacteria, thus creating distinct clusters by diet and DSS treatment (Figure 6C). Subsequent LEfSe analysis revealed several strains with significantly different relative abundance among groups and emphasized possible associations between disease phenotype and gut microbiota (Figure 6E). While SD was mainly associated with class Bacilli, order Bacteroidales, and a few members of Firmicutes phylum, CD consumption resulted in an increase in family Clostridiaceae. Subsequent induction of colitis in CD group led to an increase in *Mucispirillum schaedleri* and genus *Desulfovibrio*. Interesting shifts were observed in the HFiD group, where genera *Oscillospira* and *Akkermansia* were significantly enriched. Inflammation induction in the HFiD group led to increased abundance of genus *Anaerotruncus*, a butyrate-producing strain that could help to alleviate colitis in this group. In the HSD group, we observed an enrichment of family Enterococcaceae and genus *Turicibacter*. Colitis induction then increased the relative abundance of *Escherichia coli* from 1% in healthy mice to 23.9% in colitic mice. Metabolic pathways predicted from relative abundances revealed changes mostly associated with cell membrane transporters which were enriched in colitic HSD mice and reduced in HFiD (Appendix A).

Fungal strain dysbiosis was well observed on alpha diversity indices. In contrast to bacteria, diet switch led to an increase in observed species and their abundance, whereas administration of DSS markedly reduced both indices in all mice (Figure 6B). Diet change as well as colitis shifted fungal abundances among samples, which were shown on Bray–Curtis PCoA plots, and similar to bacteria, mice clustered together according to their treatment. Nevertheless, DSS administration led to more pronounced changes in fungal microbiota (Figure 6D). LEfSe analysis revealed several genera, such as *Aspergillus*, *Claviceps*, *Vishniacozyma*, *Alternaria*, *Xeromyces*, and *Diaporthe* that were more abundant in the group consuming SD. CD and HFiD promoted the relative abundance of *Penicillium bialowiezense* and genus *Mucor*, respectively. Both strains were probably acquired as food contaminants. Interestingly, in the HSD group, colitis led to a significant increase of *Candida* genus from 3% in healthy mice to 95.7% in colitic mice (Figure 6F).

### 3.6. Diet Significantly Influences Fecal Metabolic Profile

NMR-based metabolomics was used to follow metabolic changes in fecal samples induced by a different type of diet. The spectra contained a substantial background of broad resonances, complicating the analysis by their overlap with small metabolites signals. Therefore, intensive baseline correction was necessary to improve the spectra’s quality and to allow their reliable evaluation. PCA score plots displayed significant trends in sample clustering and detected separation of HSD and HFiD groups without any outliers (Figure 7A). The PLS-DA was executed to determine the spectral regions responsible for the differences between mice consuming different diets. However, a rather poor model validation may indicate the potential risk of overfitting (PLS-DA data not shown). Therefore, the evaluation of metabolic changes was based on univariate statistics.

A total of 157 bins created by intelligent bucketing were analyzed using parametric ANOVA. Signals in the spectral regions responsible for the groups’ differentiation were assigned to metabolites. Significant changes in normalized concentrations of 17 metabolites were revealed, namely, amino acids (alanine, aspartate, glutamate, isoleucine, lysine, methionine, phenylalanine, and tyrosine), short-chain fatty acids (acetate and propionate), and also glucose, lactate, fumarate, bile acids, uracil, methanol, and fatty acids (Table 1).

The comparison of HSD and CD groups showed that the abundance of simple saccharides in the diet affected only a few metabolites. The main change, distinguishing the HSD group from CD and HFiD, was the considerable increase in broad lipids signals (Figure 7B), detectable simply by the spectra’s visual inspection. A significant increase in methionine and decrease in glucose and methanol levels were also observed compared with the CD group. On the other hand, the effect of high fiber in the diet was more pronounced. The main difference between CD and HFiD groups was the significantly raised level of SCFAs (Figure 7B). Butyrate, not listed in Table 1, was overlapped in proton spectra by the broad fatty acid signals and therefore was not quantified. However, its increase in the HFiD group was visible in 1D projections of *J*-resolved spectra and was analogous to acetate and propionate changes. In addition, an increase in fumarate and decreased levels of glucose, lactate, bile acids, uracil, lysine, tyrosine, and aspartate were observed in the HFiD group compared with the CD group. The most significant difference was observed between HFiD and HSD groups, which was also confirmed by the PCA score plots (Figure 7A). Except for glucose, fumarate, and bile acids, significant differences were detected in all metabolites listed in Table 1. The levels of most of these metabolites decreased; only the levels of SCFAs and methanol increased.

### 3.7. HSD Detrimental Effect Is Not Transferred by Microbiota Alone

To observe whether the detrimental effect of HSD was transferred by microbiota, we colonized GF mice with microbiota (exGF) from healthy donor mice kept on CD, HSD, or HFiD. Colitis was equally severe and no significant differences in weight loss, colon length, and disease activity score were observed (Figure 8A for HSD and Appendix A for HFiD). Nevertheless, in mice colonized with HSD microbiota, the proportions of colonic neutrophils were markedly increased, although not significantly (Figure 8B), suggesting their enhanced response to altered gut microbiota from HSD-fed mice during colitis. Taken together, diet–microbiota interaction was important for both the deleterious effect of HSD and beneficial effect of HFiD.

### 3.8. Pro-Inflammatory Effect of HSD Is Abrogated in TLR4-Deficient Mice and Is Not Limited to the Gut

Our results suggested that microbiota was involved in the HSD detrimental effect. Thus, we induced colitis in TLR4-deficient mice with impaired microbial sensing (healthy mice are shown in Appendix A). Colitis in HSD-fed TLR4-deficient mice was markedly milder and almost equally severe as in CD-fed mice. Weight loss during DSS colitis in CD- or HSD-fed TLR4-deficient mice was not as pronounced as in wild-type (WT) mice. Disease activity score and colon shortening in HSD-fed TLR4-deficient mice were significantly different when compared with their wild-type counterparts (*p* < 0.0001 for both parameters; Figure 9).

Moreover, we tested whether HSD-induced pro-inflammatory tuning was strictly limited to the gut. For this purpose, we injected λ-carrageenan into the footpads of CD- or HSD-fed mice and 24 h later, we measured footpad swelling. While footpad swelling in HSD-fed wild-type mice was non-significantly bigger, in TLR4-deficient mice, no increase was observed (Appendix A). In addition, LBP serum concentration in TLR4-deficient mice was lower than that of wild-type mice regardless of the diet (Appendix A). These experiments suggested that TLR4 cascade was important in the mechanism of HSD-induced pro-inflammatory changes and that the effect of HSD was not limited to the gut.

## 4. Discussion

Diet has a major effect on gut microbiota and its components, and gut microbiota has a fundamental effect on immune system development and regulation [15,16]. These changes then drive the reactivity of the immune system, but also predispose individuals to numerous metabolic, autoimmune, and neoplastic diseases [49,50,51,52]. Here, we demonstrated that a sucrose-enriched diet impairs gut barrier function, increases inflammatory tuning of the immune system, and predisposes to intestinal inflammation. This effect is not directly transferred by microbiota alone, but is dependent on TLR4 signaling.

The western diet, which is characterized by consumption of great amounts of fats and simple refined sugars, has been associated with persistent mild systemic inflammation [52]. In our experiments, we found that compared with CD- and HFiD-fed mice, HSD-fed healthy mice had significantly increased gut permeability. This is in agreement with other studies where diets high in simple sugars (e.g., sucrose, fructose, or glucose) impaired gut barrier function and increased intestinal permeability [15,16,19]. Moreover, HSD-consuming healthy mice showed increased spleen weight and immune cells redistribution which is a marker of systemic immune system activation. This increase could be attributed to greater bacterial translocation across the impaired intestinal barrier [53]. In addition, we observed that HSD consumption increased pro-inflammatory cytokines, such as IL-1β and TNF-α in spleen and colon. Indeed, some of the animal studies on sugar effects have revealed, besides increased permeability, pro-inflammatory tuning of the organism expressed with pro-inflammatory cytokines or endotoxemia [17,18]. Our results suggested that HSD induces a state of mild sub-clinical inflammation by increasing the translocation of microbial components due to gut barrier failure.

Since we found that gut barrier was compromised even in healthy mice, we induced acute DSS colitis in HSD-fed mice and found that they were more sensitive to acute intestinal inflammation when compared with CD- or HFiD-fed mice. Colitis in HSD-fed mice was associated with pronounced weight reduction and infiltration of neutrophils into colon, together with increased levels of pro-inflammatory cytokines. Increased susceptibility to colitis has been also observed in other studies in mice fed with high-fructose [18,32,33,34], high-glucose [18,34], or high-sucrose diets [17,34]. Unfortunately, none of these studies followed the effect of HSD in the model of chronic colitis, which allows to study repeated remissions and relapses typical for ulcerative colitis [54]. In this model, we observed the protective effect of HFiD, which may be caused by the anti-inflammatory effect of SCFAs. When compared with HSD, there were less neutrophils and monocytes and lower production of pro-inflammatory cytokines in the colon of HFiD-fed mice. Interestingly, enrichment of regulatory T cells is often presented as the main anti-inflammatory mechanism of SCFAs in the gut [20]. Nevertheless, we did not observe it and even RAG2^−/−^ mice, that is, mice without T cells, showed a beneficial effect of HFiD feeding. In mice fed with HSD, proportions of neutrophils and monocytes stayed high as well as their production of iNOS, suggesting severe inflammation. Altogether, HSD feeding influenced the severity of colitis and the severe course of inflammation was even more pronounced in the case of chronic colitis and HFiD decreased the severity of chronic colitis independently of regulatory T cells.

Intact gut barrier is a hallmark of healthy mucosal immune system [21]. In our experiments, we observed impaired mucosal immune response; hence, we further analyzed gut barrier function in the colon. We found that the expression of *Muc2* was reduced, while *Il22*, *Reg3b*, and *Reg3g* were increased in healthy HSD-fed mice. Mucin 2 is the main protein of mucus layer and, together with other components, forms a glycoprotein network, preventing microbiota from direct contact with intestinal epithelium [55]. Muc2^−/−^ mice, which have increased gut permeability and high levels of Gram-negative bacteria in spleen and lymph nodes, are more prone to sepsis when challenged with a low dose of LPS [53]. Recent studies have shown that consumption of a high-fructose diet causes a reduction of mucus thickness, which results in close proximity of microbiota to intestinal epithelial cells [33,34]. Upon stimulation with microbes, epithelial cells produce greater amount of antimicrobial peptides Reg3β and Reg3γ [56]. This production is mostly stimulated by IL-22 [57]. Recently, it has been proposed that IL-23 promotes Reg3β production by epithelial cells and that Reg3β further recruits neutrophils producing IL-22 [58]. IL-22 is expressed continuously in the small intestine, but the expression in colon is very low and increases in the case of inflammation [56]. IL-22 is secreted by various immune cells, but neutrophils, in particular, have been shown to secrete IL-22 in response to DSS-induced colitis [59]. In our experiments, we observed an increase in *Il22* expression, together with higher infiltration of neutrophils, in inflamed colon of HSD-fed mice. Thus, although IL-22 maintains gut barrier integrity by stimulating the production of mucin, antimicrobial peptides, and tight junction proteins in healthy gut [60], during colitis it is overproduced especially in HSD-fed mice and could contribute to greater damage of the mucosa. It has been also shown that uncontrolled secretion of IL-22 promotes inflammation [61]. Moreover, increased levels of IL-22 have been observed in IBD patients [56]. Taken together, our results suggested that HSD enhanced immune response to DSS colitis via increased microbiota translocation, neutrophil accumulation, and related *Il22* overexpression in the colon.

Diet has been shown to shape the gut microbiota according to main dietary components [62]. In our experiments, in HFiD-fed mice, gut microbiota sequencing revealed enrichment of mucin- and fiber-utilizing bacteria, such as *Oscillospira* and *Akkermansia* before colitis and *Anaerotruncus* during colitis. These genera have been broadly accepted as healthy intestine-associated strains utilizing host mucins or producing SCFAs and their reduction has been linked with IBD [63,64]. Especially, *Akkermansia* has been thoroughly studied and shown to reduce low-grade inflammation and consequent disorders, such as insulin resistance, body weight gain, or endotoxemia [65]. However, in a genetically susceptible host, *Akkermansia* can promote gut inflammation [66]. In our study, the pro-inflammatory response in HFiD-fed mice was markedly reduced, which is in agreement with broadly accepted consensus. In mice consuming HSD, we found an enrichment of genus *Turicibacter*. Interestingly, these bacteria belong to family Erysipelotrichaceae, which has been associated with severe colitis in Nlrp12^−/−^ mice [67]. Non-inflamed gut mucosa was also colonized more by family Enterococcaceae, which has been shown to bloom together with *Candida* outgrowth [68]. Indeed, further changes in microbiota that were associated with DSS administration revealed increased abundance of *E. coli* and genus *Candida*. These microbes have been linked to gut inflammation in IBD patients as well as in animal models [69,70]. Recently, Sovran and colleagues have shown that *Escherichia* promotes *Candida* growth and that, together they exacerbate colitis [71]. Taken together, HSD influences gut microbiota composition and function and promotes the growth of potentially pathogenic bacteria and fungi.

The fecal NMR metabolomics results showed no significant difference between CD and HSD, but HFiD implicated important changes. As expected, we observed an increased level of SCFAs, end products of catabolism of polysaccharides, and sugars. They are crucial energy sources for colonic epithelial cells, but also can influence the host immune system [72,73]. We observed decreased lactate in HFiD, which is a product of lactic acid bacteria that is not usually detected in feces under normal conditions [73,74]. Its increased levels have been detected in patients with active ulcerative colitis [75,76]. Decreased lactate level can be attributed to the presence of microbiota utilizing it for SCFAs [74,75,77]. In addition, we detected a decrease in the concentration of amino acids when comparing HFiD with HSD, which agrees with the results of Smith and Macfarlane [78]. Increased amino acid uptake by intestinal bacteria in the presence of fermentable carbohydrates and simultaneous decreased ammonia formation have been explained by higher biosynthetic activity of intestinal bacteria [78,79]. These data suggest that supplementation with polysaccharides can shape the microbiome and probably upregulate its biosynthetic pathways.

A marked increase in fatty acid concentration was demonstrated in HSD. Recently, high-fat diet and particularly, fatty acids have been shown to trigger the TLR4 cascade, resulting in inflammation [80,81]. We found that preserved TLR4 signaling was important for severe colitis induction in HSD-fed mice as TLR4-deficient mice showed milder signs of colitis when compared with wild-type mice. This effect has been also observed after lipid infusion where TLR4 knockout markedly reduced pro-inflammatory response in fatty tissue [82]. Interestingly, in the colon of wild-type mice, we did not find any changes in *Tlr4* expression, but negative regulators of the TLR4 pathway NF-kappa-B inhibitor zeta and interleukin-1 receptor-associated kinase 3 (IRAK3) showed increased expression. It has been previously shown that in case of prolonged TLR4 stimulation, IRAK3 works to turn down TLR expression in order to restore homeostasis and prevent more damage caused by TLR pathway-induced pro-inflammatory cytokines [83,84]. These findings raised a question of whether dysbiotic microbiota of HSD-fed mice can induce any pro-inflammatory changes. Therefore, we transferred intestinal contents of mice that were fed different diets into germ-free recipients. This experiment revealed that HSD microbiota induced immune cell redistribution, but did not aggravate DSS-induced colitis. This is an interesting finding because we transferred high-sucrose-associated microbiota, while two other studies using microbiota from high-fructose- or high-glucose-fed mice showed increased severity of colitis in exGF mice [33,34].

In conclusion, we have shown that high consumption of monosaccharides and disaccharides increases pro-inflammatory tuning of the organism, compromises the gut barrier function, and worsens colitis in a TLR4-dependent manner. While severe colitis is associated with clear gut microbiota dysbiosis with increased abundance of *E. coli* and *Candida*, the phenotype is not transferred to germ-free recipients. Mucin reduction and subsequent translocation of *E. coli*-derived LPS or access of lipidic metabolites trigger TLR4 signaling pathway and induce pro-inflammatory cytokine production and immune cell redistribution, finally resulting in systemic low-grade inflammation. Such a setting probably accelerates the immune response, resulting in severe DSS colitis and even promoting inflammation outside the gut.

## Figures and Tables

**Figure 1 cells-09-02701-f001:**
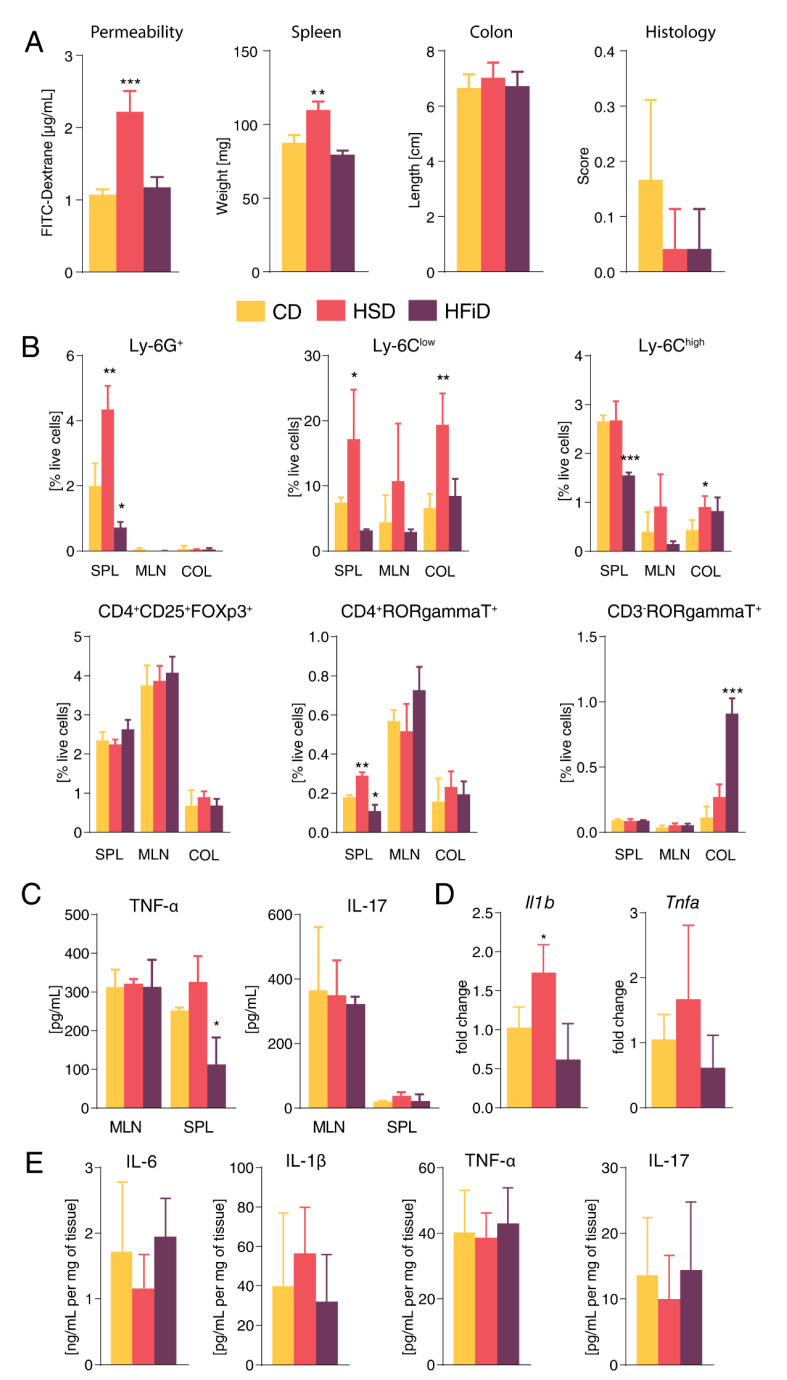
Diet rich in simple sugars significantly increases intestinal permeability and promotes local and systemic pro-inflammatory tuning. (**A**) Intestinal permeability, spleen weight, colon length, and histological score of mice consuming CD, HSD, and HFiD. (**B**) Percentage of live cells from spleens, mesenteric lymph nodes, and colons measured by flow cytometry. (**C**) TNF-α and IL-17 production in cell suspensions from mesenteric lymph nodes and spleens stimulated with anti-CD3/anti-CD28 antibodies was measured by ELISA. (**D**) The mRNA expression of pro-inflammatory cytokines in colon. (**E**) Cytokine levels measured in supernatants from colonic tissue cultures by ELISA. Data presented are from one representative experiment out of three, *n* = 3 mice per group. Means significantly different from CD were calculated by one-way analysis of variance (ANOVA) with Dunnett’s multiple comparison test in a respective organ. Data are shown as mean ± SD. * *p* < 0.05; ** *p* < 0.01; *** *p* < 0.001. CD, control diet; HSD, diet rich in simple carbohydrates; HFiD, diet rich in fiber; SPL, spleen; MLN, mesenteric lymph nodes; COL, colon.

**Figure 2 cells-09-02701-f002:**
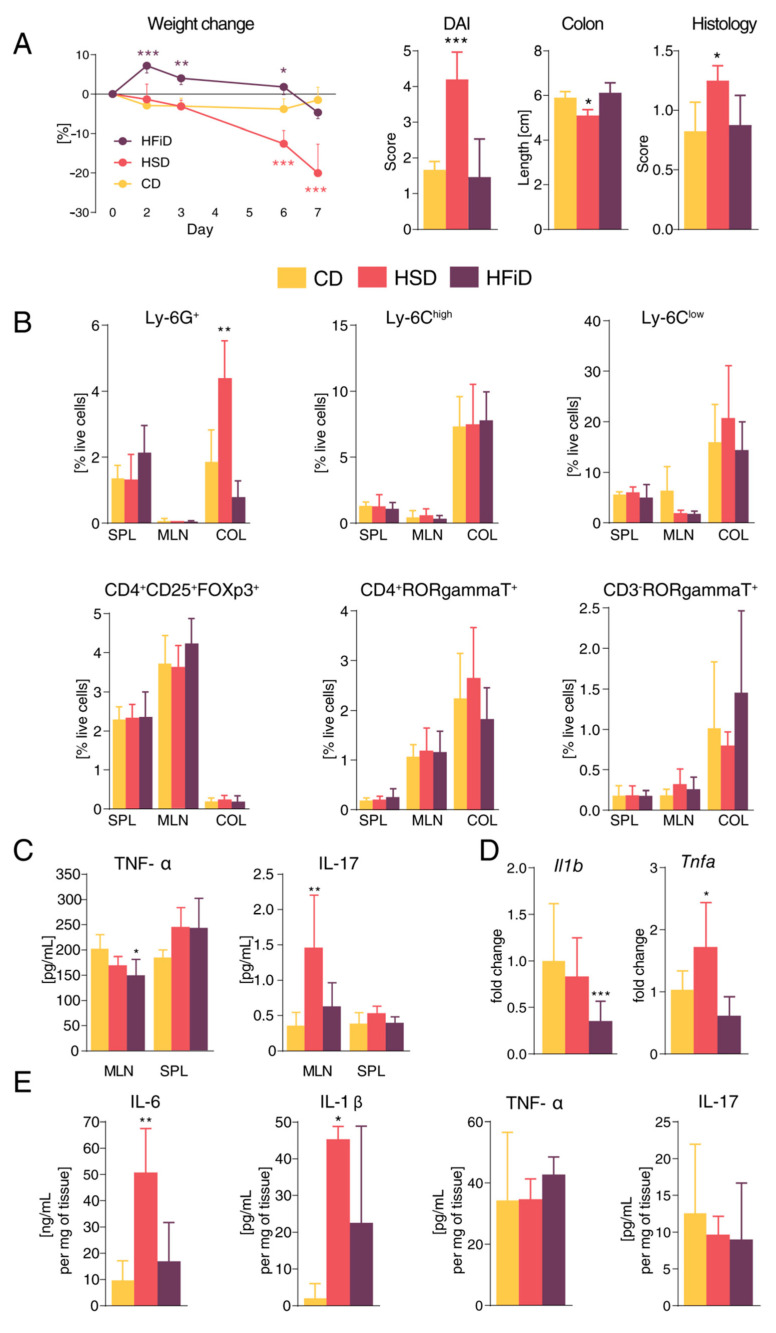
Simple sugars significantly modify colitis course and mucosal immune response. (**A**) Weight change during dextran sodium sulfate treatment, final disease activity score (DAI), colon length, and histology score. (**B**) Percentage of live cells from spleens, mesenteric lymph nodes, and colons measured by flow cytometry. (**C**) TNF-α and IL-17 production in cell suspensions from mesenteric lymph nodes and spleens stimulated with anti-CD3/anti-CD28 antibodies. (**D**) The mRNA expression of pro-inflammatory cytokines in colon. (**E**) Cytokines measured in tissue culture of colon biopsies from diseased mice. Data presented are from one representative experiment out of three, *n* = 5–8 mice per group. Means significantly different from CD were calculated. The weight loss was analyzed by two-way ANOVA with Bonferroni post hoc test and other parameters by one-way ANOVA with Dunnett’s multiple comparison test in a respective organ. Data are shown as mean ± SD. * *p* < 0.05; ** *p* < 0.01; *** *p* < 0.001. CD, control diet; HSD, diet rich in simple carbohydrates; HFiD, diet rich in fiber; SPL, spleen; MLN, mesenteric lymph nodes; COL, colon.

**Figure 3 cells-09-02701-f003:**
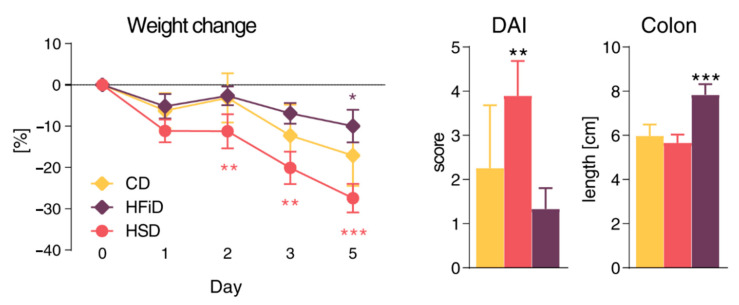
Simple sugars significantly worsen colitis course in RAG2^−/−^ mice. Weight reduction, final disease activity score (DAI), and colon length of mice after DSS colitis induction. Data presented are from two experiments, *n* = 5–9 mice per group. Means significantly different from CD were calculated. The weight loss was analyzed by two-way ANOVA with Bonferroni post hoc test and other parameters by one-way ANOVA with Dunnett’s multiple comparison test. Data are shown as mean ± SD. * *p* < 0.05; ** *p* < 0.01; *** *p* < 0.001. CD, control diet; HSD, diet rich in simple carbohydrates; HFiD, diet rich in fiber.

**Figure 4 cells-09-02701-f004:**
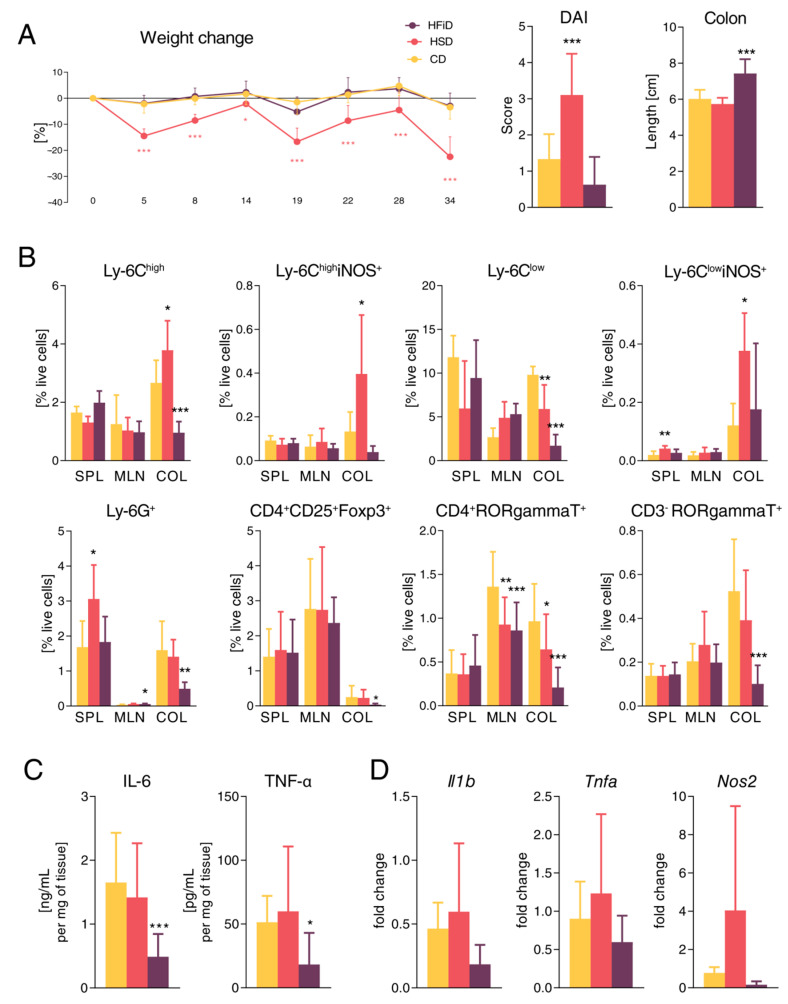
HSD significantly increases the severity of chronic colitis. (**A**) Weight loss, disease activity score, (DAI) and colon length. (**B**) Percentage of live cells from spleens, mesenteric lymph nodes, and colons measured by flow cytometry. (**C**) Cytokines measured in tissue culture of colon biopsies from diseased mice. (**D**) The mRNA expression of pro-inflammatory cytokines and *Nos2* in colon. Data presented are from one representative experiment out of two, *n* = 8 mice per group. Means significantly different from CD were calculated. The weight loss was analyzed by two-way ANOVA with Bonferroni post hoc test and other parameters by one-way ANOVA with Dunnett’s multiple comparison test in a respective organ. Data are shown as mean ± SD. * *p* < 0.05; ** *p* < 0.01; *** *p* < 0.001. CD, control diet; HSD, diet rich in simple carbohydrates; HFiD, diet rich in fiber; SPL, spleen; MLN, mesenteric lymph nodes; COL, colon.

**Figure 5 cells-09-02701-f005:**
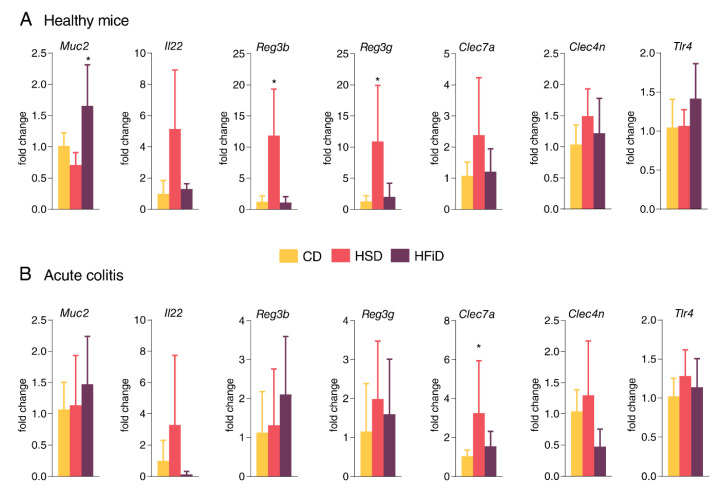
Colonic mRNA expression of genes related to gut barrier was changed. (**A**) HSD significantly influenced mRNA expression of antimicrobial peptides *Reg3b, Reg3g*, and *Il22* (not significant) and slightly influenced mRNA expression of pattern recognition receptor dectin 1 (*Clec7a*). (**B**) DSS-induced acute colitis had partial influence on *Reg3b* and *Reg3g* mRNA expression, but all other measured genes showed a similar trend as in healthy mice. Data presented are from one experiment out of three, *n* = 5–8 mice per group. Means significantly different from CD were calculated by one-way ANOVA with Dunnett’s multiple comparison test. Data are shown as mean ± SD. * *p* < 0.05. CD, control diet; HSD, diet rich in simple carbohydrates; HFiD, diet rich in fiber.

**Figure 6 cells-09-02701-f006:**
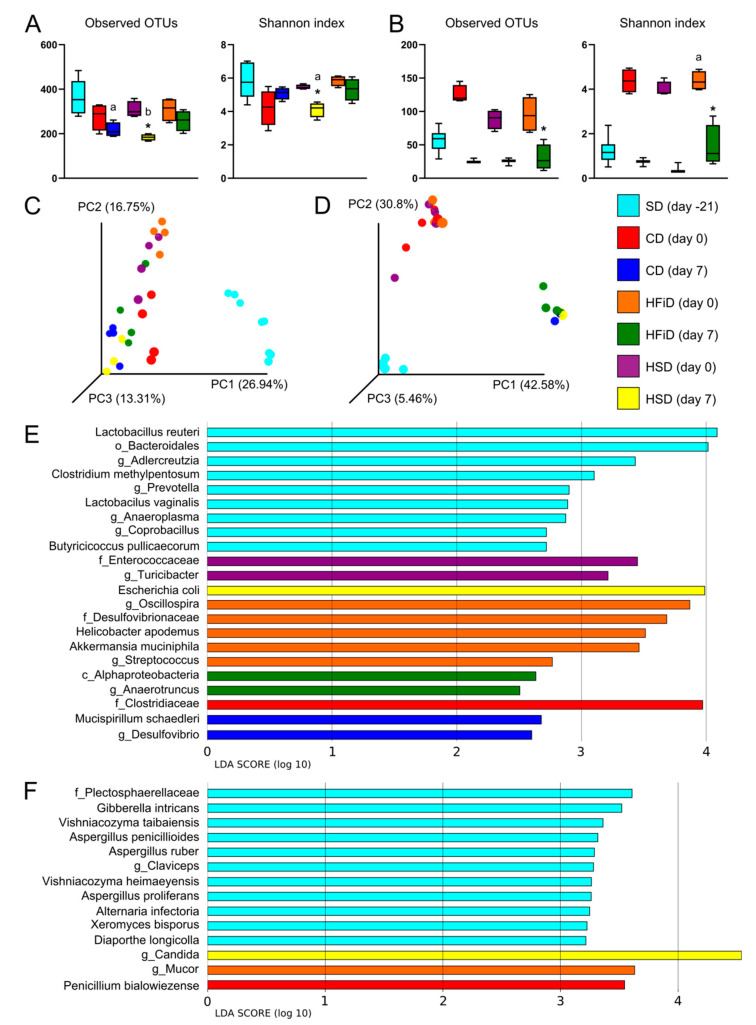
Diet and DSS treatment influence gut microbiota composition. Alpha diversity indices: observed operational taxonomic units (OTUs) and Shannon index for (**A**) bacteria and (**B**) fungi. The data are presented as box plots with determined mean and standard deviations (three or four mice per group). Means significantly different from day-21 (SD = standard diet) were calculated by ANOVA with *p* < 0.05 (a) and *p* < 0.01 (b), and means significantly different from respective day 0 (i.e., healthy mice vs. mice with colitis on day 7) were analyzed by *t* test * (*p* < 0.05). Beta diversity results: Bray–Curtis principal co-ordinate analysis (PCoA) plots for (**C**) bacteria and (**D**) fungi and linear discriminant analysis effect size (LEfSe) analyses of significantly different strains of (**E**) bacteria and (**F**) fungi among groups. CD, control diet; HSD, diet rich in simple carbohydrates; HFiD, diet rich in fiber.

**Figure 7 cells-09-02701-f007:**
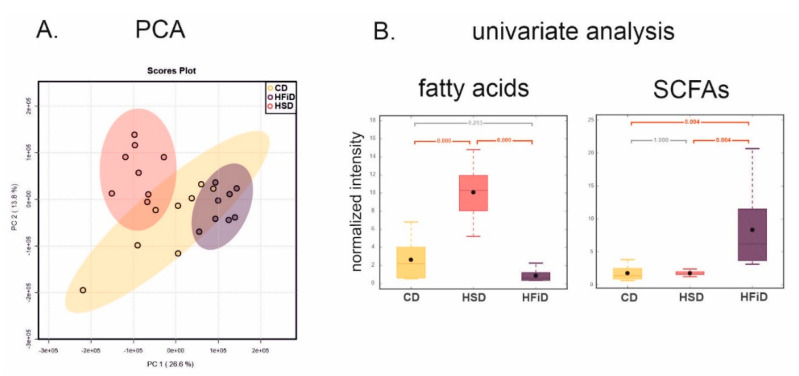
Diet-induced changes in the fecal metabolome. (**A**) Score plots of principal component analysis (PCA) model. (**B**) Box plots reflecting levels of short-chain fatty acids (SCFAs) and fatty acids. *p*-Values in box plots were calculated by parametric ANOVA. CD, control diet; HSD, diet rich in simple carbohydrates; HFiD, diet rich in fiber.

**Figure 8 cells-09-02701-f008:**
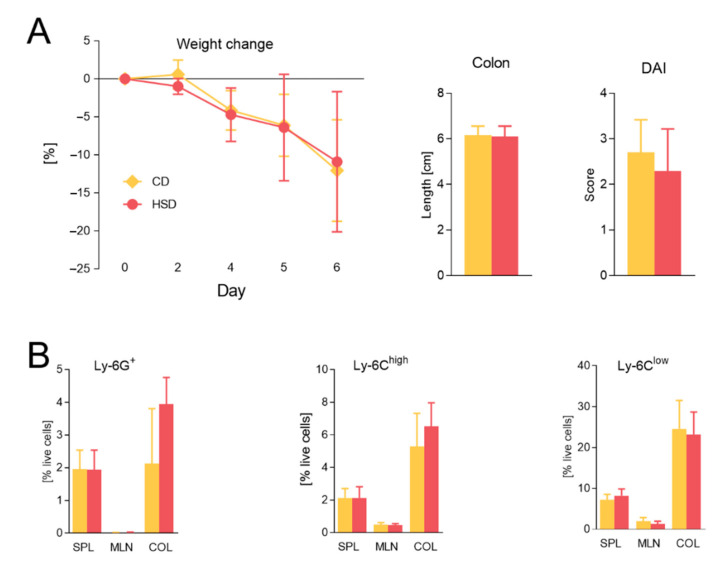
Harmful effect of HSD in acute colitis is not transferred by microbiota alone. (**A**) Weight change, colon length, and DAI are shown. (**B**) Percentage of live cells from spleens, mesenteric lymph nodes, and colons measured by flow cytometry. Data presented are from two experiments, *n* = 5–8 mice per group. The weight loss was analyzed by two-way ANOVA with Bonferroni post hoc test and other parameters by unpaired *t* test in a respective organ. Data are shown as mean ± SD. CD, control diet; HSD, diet rich in simple carbohydrates; SPL, spleen; MLN, mesenteric lymph nodes; COL, colon.

**Figure 9 cells-09-02701-f009:**
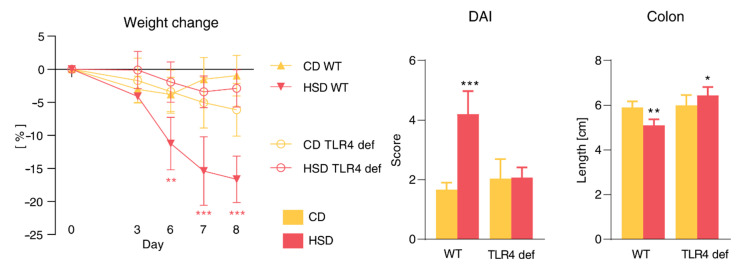
Detrimental effect of HSD is abrogated in TLR4-deficient mice. Weight change, DAI, and colon length. Data presented are from two experiments, *n* = 5–8 mice per group. The weight change was analyzed by two-way ANOVA with Bonferroni post hoc test and other parameters by unpaired *t* test in a particular genotype. Data are shown as mean ± SD. * *p* < 0.05, ** *p* < 0.01, *** *p* < 0.001. CD, control diet; HSD, diet rich in simple carbohydrates; WT, wild-type mice; TLR4 def, TLR4-deficient mice.

**Table 1 cells-09-02701-t001:** Significant diet-induced changes in fecal metabolites concentrations.

	HFiD vs. CD		HSD vs. CD		HFiD vs. HSD	
	Δ (%)	*p*-Value	∆ (%)	*p*-Value	∆ (%)	*p*-Value
SCFAs						
Acetate	**369.74**	**0.0056**	−1.83	0.9998	**378.48**	**0.0054**
Propionate	**461.72**	**0.0002**	13.85	0.9885	**393.38**	**0.0004**
Amino acids						
Alanine	−31.70	0.2688	47.74	0.0631	**−** **53.77**	**0.0018**
Aspartate	**−** **51.09**	**0.0107**	22.51	0.3469	**−** **60.08**	**0.0004**
Glutamate	−35.44	0.1888	34.15	0.2108	**−** **51.88**	**0.0050**
Isoleucine	−22.23	0.6574	46.09	0.1849	**−** **46.77**	**0.0337**
Lysine	**−** **50.35**	**0.0485**	14.14	0.7591	**−** **56.50**	**0.0103**
Methionine	−43.22	0.0935	**55.89**	**0.0248**	**−** **63.58**	**0.0001**
Phenylalanine	−26.49	0.3116	40.13	0.0824	**−** **47.54**	**0.0031**
Tyrosine	**−** **55.35**	**0.0191**	30.80	0.2460	**−** **65.86**	**0.0004**
Others						
Glucose	**−** **66.04**	**0.0000**	**−** **48.78**	**0.0006**	−33.70	0.2779
Lactate	**−** **40.35**	**0.0074**	−9.82	0.6917	**−** **33.85**	**0.0455**
Fumarate	**47.72**	**0.0195**	9.66	0.8221	34.72	0.0688
Bile acids ^a^	**−** **67.97**	**0.0206**	−34.37	0.3179	−51.19	0.3336
Methanol	4.45	0.7509	**−** **16.25**	**0.0383**	**24.71**	**0.0077**
Uracil	**−** **62.30**	**0.0043**	16.47	0.6092	**−** **67.63**	**0.0004**
Fatty acids	−67.10	0.2928	**283.60**	**0.0000**	**−** **91.42**	**0.0000**

Data are expressed as percentage change of normalized concentrations of HFiD/CD, HSD/CD, and HFiD/HSD; significant changes are marked in bold. *p*-Values were calculated by parametric ANOVA. SCFAs, short-chain fatty acids; CD, control diet; HSD, diet rich in simple carbohydrates; HFiD, diet rich in fiber; ^a^ tentative assignment.

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
