# Peer review of "Diet Rich in Simple Sugars Promotes Pro-Inflammatory Response via Gut Microbiota Alteration and TLR4 Signaling"

_cells, 2020, doi:10.3390/cells9122701_

Round 1

Reviewer 1 Report

The manuscript by Fajstova et al, entitled « Diet rich in simple sugars promotes pro-inflammatory response via gut microbiota  alteration and TLR4 signaling », aims to decipher the impact of diet enriched in simple sugar (HSD), in comparison to diet enriched in fibers (HFiD), on gut permeability, modulation of the immune response, susceptibility to colitis and changes in the gut microbiota composition. Even if most experimental studies studying the impact of ‘western diet’ in IBD have focused on high-fat rich diet, recent studies have already highlighted that high-sugar diet resulted in the deterioration of intestinal barrier function and increased susceptibility to DSS-induced colitis. Clinical trials in humans have also demonstrated an association between intake of refined sugar and increased incidence of IBD which can be reversed by a fiber enriched diet, notably thought the production of short-chain fatty acids (SCFAs) by specific bacterial populations. So for those reasons, this work is not really original. However, this study brings new insights and some mechanisms, notably the impact on immune responses, on gut microbiota and metabolome and the role of TLR4 signaling. They also performed fecal transfer to germ free mice. However some controls are missing. Altogether the study is well conducted, but they are some concerns that the authors should take into account to improve the manuscript.

Major comments

  • In the experiment in healthy mice (Figure 1A), the authors showed that basal permeability is increased by sugar enriched diet, but the level of FITC-dextran in control healthy mice is of 10 µg/ml, while when they repeat the experiment with RAG-/-  mice, the basal level is 10 time less (1µg/ml).  How the authors explain that? What represent the histological score in healthy mice???
  • In DSS colitis (Figure 2), it miss the control group of healthy mice (without DSS) in order to compare the scores levels and impact on immune response in healthy mice (CD without DSS), CD with DSS/ HSD DSS/ HiFD DSS. Indeed, it seems that the control group with DSS (CD) presents a very low colitis with no weight loss, and it is impossible to compare the DAI, shortening of the colon, histology and immune responses from the basal level.

Again for qPCR, the normalization has been done with control mice with colitis (control CD-DSS) and not with healthy mice. Idem for cell phenotyping, it is impossible to have the basal level and to follow the impact of colitis induction

  • Idem in Figure 3, it would have been important to compare the impact of DSS in RAG 2-/- to healthy mice and to WT mice with DSS
  • Idem in chronic colitis, the control group of healthy mice is missing. It seems that in DSS control group (CD), no colitis is observed since there is no weight loss and the DAI is 1. Again without control healthy mice, it is impossible to evaluate the colon shortening and the impact of DSS on the development of inflammation and deregulation of immune responses
  • HSD increases the expression IL-22 and antimicrobial peptides and occludin. These markers could also favor the gut barrier integrity rather than permeability. The authors explain should better discuss the role of neutrophils versus epithelial cells.
  • P16, line435: the raised level of SCFA is observed for HFiD and not for HSD as indicated in the text
  • P18, line 455: The authors indicated that GF mice were colonized with microbiota from healthy mice on CD, HSD and HFiD, but the graph (Figure 8) did not show the results with mice colonized with microbiota from HFiD mice. The authors should mention that they perfomed a colitis (DSS / dose…) after colonization and precise the number of days after colonization. This is not indicated in the M&M. How the authors explain that the phenotype of colitis susceptibility was not transferred? Concerning the changes in cellular populations, they are not statistically different.
  • The results obtained in TLR4-/- are very interesting, however as previously the control healthy mice is missing.

The discussion is quite interesting and discussed all the results obtained in comparison to what have been already reported in the literature.

Minor comments

P2, line 62: SCFA… can influence the recruitment of colonic T regulatory instead of the counts

P2, line 68: then instead of than

P2, line 77: The influence of wester diet has been recognized…, including instead of most of them

P3, line 95: standard diet instead of maintenance diet

P3, line 103: indicate the dilution factor of the intestinal content for fecal transfer

Are the authors sure that the volume indicated for the rectal administration (200 µl) is correct, since most of the liquid should have gone out?

P3, line 111-115: Usually for FITC-Dextran permeability assay, mice should have been fasted. Precise if this was the case

P4, line 160: at a concentration of …

P4, line 164-165: please provide more detail on the standardization of the protocol (weight of colon samples/ ml?)

P6, line 257: Diet enriched in simple carbohydrates or sugar-rich diet

Figure 1: idem: simple sugar enriched diet

P9, line 277: TNF-a and IL-17 production was measured by ELISA

P9, line 278: mRNA expression

P9, line 299: Even if instead of while .., in both healthy and colitic conditions

P9, line 301: more severe colitis (colitis is missing)

P13, line 351: Since HDS induced changes, .instead of since we observed changes

P19, line 496-497: diet rich in simple sugar or sucrose-enriched

Author Response

We thank the reviewer for her/his suggestions and comments. We responded to all of them and where accepted we made changes in the manuscript.

Comments and Suggestions for Authors

The manuscript by Fajstova et al, entitled « Diet rich in simple sugars promotes pro-inflammatory response via gut microbiota  alteration and TLR4 signaling », aims to decipher the impact of diet enriched in simple sugar (HSD), in comparison to diet enriched in fibers (HFiD), on gut permeability, modulation of the immune response, susceptibility to colitis and changes in the gut microbiota composition. Even if most experimental studies studying the impact of ‘western diet’ in IBD have focused on high-fat rich diet, recent studies have already highlighted that high-sugar diet resulted in the deterioration of intestinal barrier function and increased susceptibility to DSS-induced colitis. Clinical trials in humans have also demonstrated an association between intake of refined sugar and increased incidence of IBD which can be reversed by a fiber enriched diet, notably thought the production of short-chain fatty acids (SCFAs) by specific bacterial populations. So for those reasons, this work is not really original. However, this study brings new insights and some mechanisms, notably the impact on immune responses, on gut microbiota and metabolome and the role of TLR4 signaling. They also performed fecal transfer to germ free mice. However some controls are missing. Altogether the study is well conducted, but they are some concerns that the authors should take into account to improve the manuscript.

Major comments

  • In the experiment in healthy mice (Figure 1A), the authors showed that basal permeability is increased by sugar enriched diet, but the level of FITC-dextran in control healthy mice is of 10 µg/ml, while when they repeat the experiment with RAG-/-  mice, the basal level is 10 time less (1µg/ml).  How the authors explain that? What represent the histological score in healthy mice???

Answer: Thank you for this comment. We apologize for this, we used an older version of the figure 1A with older dataset with different calculation of FITC-dextran level. Now, we used newer data and correct version of the image in Figure 1A.

Thank you for this question. Histological score quantitatively evaluates the damage of the mucosa including infiltration with leucocytes. It may be increased in healthy mice when barrier damage is associated with leucocytes infiltration. It is used for a long time in our laboratory and thus we forgot to specify the way of histological samples evaluation, so we added this to M&M section on line 129.

  • In DSS colitis (Figure 2), it miss the control group of healthy mice (without DSS) in order to compare the scores levels and impact on immune response in healthy mice (CD without DSS), CD with DSS/ HSD DSS/ HiFD DSS. Indeed, it seems that the control group with DSS (CD) presents a very low colitis with no weight loss, and it is impossible to compare the DAI, shortening of the colon, histology and immune responses from the basal level.

Again for qPCR, the normalization has been done with control mice with colitis (control CD-DSS) and not with healthy mice. Idem for cell phenotyping, it is impossible to have the basal level and to follow the impact of colitis induction

Answer: Thank you for these comments. The reason, we present our data this way, is that we want to highlight the changes in healthy mice (Figure 1). These mice were treated the same way, except DSS, as the other mice and were usually collected when the rest of mice got the DSS. We called them background mice. But when we found all the important parameters were changed in them we exclude these results and present them on their own. As all the parameters are comparable among Figures, reader can easily observe how they are changed after disease induction. Indeed, most of them would be significantly different when comparing healthy and colitic mice. We cannot compare the severity of colitis with healthy mice when following the effect of different diets. Colitis induction is not the variable in our experiments, the diet is. We compare the data from diseased mice with disease control, the CD with DSS in this case.

  • Idem in Figure 3, it would have been important to compare the impact of DSS in RAG 2-/- to healthy mice and to WT mice with DSS

Answer: To improve the comparability of our data, we added the colon length of healthy RAG 2-/- mice to the Figure S6. All the parameters are comparable among Figures.

  • Idem in chronic colitis, the control group of healthy mice is missing. It seems that in DSS control group (CD), no colitis is observed since there is no weight loss and the DAI is 1. Again without control healthy mice, it is impossible to evaluate the colon shortening and the impact of DSS on the development of inflammation and deregulation of immune responses

Answer: Thank you for this comment, but again healthy mice are depicted on the Figure 1. Control mice had mild colitis, with no colitis the score would be close to zero.

  • HSD increases the expression IL-22 and antimicrobial peptides and occludin. These markers could also favor the gut barrier integrity rather than permeability. The authors explain should better discuss the role of neutrophils versus epithelial cells.

Answer: Thank you for pointing out this issue. In healthy gut, IL-22 and antimicrobial peptides are produced in limited amounts by immune and epithelial cells, respectively, to improve barrier integrity. On the other hand, in colitic mice, especially after consumption of diet rich in simple carbohydrates, neutrophils infiltrate the mucosa and produce excess of IL-22 with detrimental consequences. We modified this particular paragraph in the discussion to indicate the role of IL-22 in healthy gut (lines 549-563).

  • P16, line435: the raised level of SCFA is observed for HFiD and not for HSD as indicated in the text

Answer: Thank you, we corrected this and HSD was replaced with HFiD.

  • P18, line 455: The authors indicated that GF mice were colonized with microbiota from healthy mice on CD, HSD and HFiD, but the graph (Figure 8) did not show the results with mice colonized with microbiota from HFiD mice. The authors should mention that they performed a colitis (DSS / dose…) after colonization and precise the number of days after colonization. This is not indicated in the M&M. How the authors explain that the phenotype of colitis susceptibility was not transferred? Concerning the changes in cellular populations, they are not statistically different.

Answer: Thank you for this comment. The graph showing the effect of HFiD microbiota transfer is in Supplementary material – Figure S9. To improve this, we specified it on line 462: “Figure 8A for HSD and Figure S9 for HFiD”. There is no beneficial effect of HFiD microbiota. The colonization length is mentioned in M&M section on line 107, but we understand that it needed to be improved, therefore we specified it by adding “…before acute DSS colitis induction.“. The concentration of DSS is mentioned on line 118 and is 3% in all experiments.

Thank you for this question. We think that, for HSD and HFiD, their actual consumption is important for the effect on immune response and colitis. This has been observed in the case of reduced SCFAs after low fiber diet consumption which do not protect against colitis. The effect of HSD metabolites was not studied so far and our study is suggesting their importance. Anyway, there are studies presenting transferability of HSD effects by microbiome, e.g. Khan et al. 2020. But there are several differences in the protocol, such as different diet (glucose enriched water) or only fecal water transfer per os.

  • The results obtained in TLR4-/- are very interesting, however as previously the control healthy mice is missing.

Answer: To show healthy TLR4 deficient mice for comparison, we added a new Figure S10 depicting the parameters of healthy TLR4 deficient mice. All the healthy mice results were similar, so we did not want to increase the numbers of mice according to 3R.

The discussion is quite interesting and discussed all the results obtained in comparison to what have been already reported in the literature.

Minor comments

P2, line 62: SCFA… can influence the recruitment of colonic T regulatory instead of the counts

Answer: Corrected, counts replaced with the word recruitment.

P2, line 68: then instead of than

Answer: Corrected, than replaced with then.

P2, line 77: The influence of wester diet has been recognized…, including instead of most of them

Answer: In our opinion, suggested adjustment changes the meaning of the sentence, therefore we did not used it.

P3, line 95: standard diet instead of maintenance diet

Answer: The “Maintenance diet” is an official title of the diet from Altromin, therefore we cannot change it.

P3, line 103: indicate the dilution factor of the intestinal content for fecal transfer

Answer: New version: “…intestinal content from colon and ileum was collected to separate tubes, pooled for all mice in respective groups and diluted in 2 mL of sterile PBS.“ We did not measured the exact volume of pooled intestinal contents from donor mice.

Are the authors sure that the volume indicated for the rectal administration (200 µl) is correct, since most of the liquid should have gone out?

Answer: Yes, although some of the liquid went out during rectal administration most of it stayed inside and assured proper colonization.

P3, line 111-115: Usually for FITC-Dextran permeability assay, mice should have been fasted. Precise if this was the case.

Answer: No, mice were not fasted. In our lab, we use this assay for more than a decade and we found the results comparable when used in non-fasted mice.

P4, line 160: at a concentration of …

Answer: Corrected.

P4, line 164-165: please provide more detail on the standardization of the protocol (weight of colon samples/ ml?)

Answer: To improve the protocol, we add these details and the sentence is now: “Colon biopsies were weighted (approximately 3-7 mg) and cultivated in 500 µL of complete RPMI medium…”.

P6, line 257: Diet enriched in simple carbohydrates or sugar-rich diet

Answer: Changed to “Diet rich in …”.

Figure 1: idem: simple sugar enriched diet

Answer: Changed to “Diet rich in simple sugars…”.

P9, line 277: TNF-a and IL-17 production was measured by ELISA

Answer: The sentence is now: “TNF-α and IL-17 production in cell suspensions from mesenteric lymph nodes and spleens stimulated with anti-CD3/anti-CD28 antibodies was measured by ELISA.”.

P9, line 278: mRNA expression

Answer: Corrected.

P9, line 299: Even if instead of while .., in both healthy and colitic conditions

Answer: The sentence is now: “Even if only minor changes in T cells were observed in HSD fed mice, in both healthy and colitic conditions, we induced acute colitis in RAG2-/- mice fed with different diets.”.

P9, line 301: more severe colitis (colitis is missing)

Answer: Colitis added.

P13, line 351: Since HDS induced changes, .instead of since we observed changes

Answer: The sentence is now: “Since HSD induced changes in intestinal permeability and mucosal immune response in healthy mice, we analyzed…”.

P19, line 496-497: diet rich in simple sugar or sucrose-enriched

Answer: The sentence is now: “Here, we demonstrate that sucrose enriched-diet impairs…”.

Reviewer 2 Report

280- Are you sure that n=3 will provide statistical significance?

304 - Would you be so kind to give  in the paper the exact value of increasing  percentage of neutrophils in spleen. My comment relates not only to this line, it is a systemic problem available at 302,329 and other lines also. You compare effects in  qualitative  characteristics, but maybe you will agree with me that it will be more comfortable for readers to learn effects in  percentage or in other quantitative characteristics for example 50% of increasing in growth of Candida etc

543-Please,  format IL22 etc appropriately 

Author Response

We thank the reviewer for her/his suggestions and comments. We responded to all of them and where accepted we made changes in the manuscript.

Comments and Suggestions for Authors

280- Are you sure that n=3 will provide statistical significance?

Answer: Thank you for this comment. The experiment was repeated with similar results therefore we believe that n=3 from one representative experiment is sufficient. We did not want to increase the numbers of mice according to 3R.

304 - Would you be so kind to give  in the paper the exact value of increasing  percentage of neutrophils in spleen. My comment relates not only to this line, it is a systemic problem available at 302,329 and other lines also. You compare effects in  qualitative  characteristics, but maybe you will agree with me that it will be more comfortable for readers to learn effects in  percentage or in other quantitative characteristics for example 50% of increasing in growth of Candida etc

Answer: Thank you for this comment. We used descriptive characteristic of the parameters because there is plenty of them changed and the exact numbers are shown on figures. We preferred clear description than enumeration of percentages. But we added the exact percentage change to E. coli and Candida (lines 392 and 406, respectively) because these are not on graphs and are important in HSD induced dysbiosis.

543-Please,  format IL22 etc appropriately 

Answer: Thank you for this comment. In this paragraph, we discuss gene expression (should be typed in italic Il22) and also protein levels or the protein in general (should be typed normally IL-22), we believe the formatting is correct in this way. If this is not what was asked by the Reviewer would you please specify what exactly should we format?

Round 2

Reviewer 1 Report

The revised version of the manuscript by Alena Fajstova et al, entitled « Diet rich in simple sugars promotes pro-inflammatory response via gut microbiota alteration and TLR4 signaling », has taken into account and replied to the main comments I made, and as such the manuscript has been largely improved. I understand that all the results concerning the effect of the different diets in naive mice were not included in the graph concerning colitis experiments, since the authors wanted to highlight changes in the parameters, however as I mentioned it would have been important to keep at least control mice (only control diet without DSS) in the graph. Indeed, this would have provided better view on the effect of DSS by itself (for both acute and chronic colitis, as well as in RAG-/- and TLR4-/- mice) since it seems that the colitis were very mild. However, effects on naïve mice have been provided separately and the effect of the HSD diet remains very clear, in comparison to colitic mice.   

The authors have implemented the discussion to respond to the vast majority of suggestions that were raised and have provided an exhaustive point-to-point rebuttal letter. These modifications have significantly improved the manuscript.

minor comments: Table 1 propionate instead of propinate